# The Impact of Air Pollution on Labor Supply in China

**Mingxuan Fan [1,\*] and Corbett Grainger [2,3,4]**

1    Department of Real Estate, NUS Business School, National University of Singapore,
     Singapore 119245, Singapore
2    Department of Agriculture and Applied Economics, University of Wisconsin-Madison,
     Madison, WI 53706, USA; corbett.grainger@wisc.edu
3    Centre for Climate and Energy Transition, Universitetet i Bergen, 5007 Bergen, Norway
4    CESifo, 81679 München, Germany
*    Correspondence: mfan@nus.edu.sg

**Abstract:** A growing body of literature demonstrates that air pollution has negative impacts on human health, cognitive ability, and labor productivity, but little is known about the effect of chronic air pollution on labor supply decisions. We use restricted-access individual-level panel data from the China Family Panel Survey (CFPS), paired with sub-district level remote-sensing estimates of air pollution, to evaluate the impact of chronic exposure to fine particulate matter ($PM_{2.5}$) on an individual's hours worked. We exploit within-individual changes in air pollution, and fixed effects estimates indicate that an increase of 1 $\mu g/m^3$ in $PM_{2.5}$ reduces an individual's average hours worked by about 14 min per week. We then leverage the city-level roll-out of air pollution monitoring and information provision to test hypotheses about the underlying mechanisms. We show that individuals with poor health respond to changes in $PM_{2.5}$ the most. For individuals who are environmentally unaware, this effect is mostly through an impact of pollution on health, while individuals who are environmentally aware engage in avoidance behavior. Finally, the roll-out of monitoring and information provision at the city level plays an important role in raising awareness and individuals' responsiveness to pollution.

**Keywords:** $PM_{2.5}$; pollution awareness; hours worked

## 1. Introduction

A large and growing body of literature has demonstrated health and mortality impacts of fine airborne particulate matter ($PM_{2.5}$). Indeed, air pollution is now considered one of the largest health threats in the world, particularly in developing countries [1]. There is strong evidence establishing a causal link between pollution and health or mortality [2,3], including recent studies focusing explicitly on mortality or morbidity in China [4–6]. Pollution has also been shown to affect school attendance [7] and cognition [8], and early exposure has been tied to human capital formation [9] and longer-run earnings [10]. It is plausible that chronic pollution exposure could also affect labor supply decisions. On the one hand, air pollution may cause short-term illnesses and affect the long-term health status, both of which lead to reduced time spent working. On the other hand, individuals respond to pollution through avoidance behavior and defensive expenditures [11–16], which not only affect the choice between labor and leisure directly, but also modify the health effect of air pollution on labor supply. Heightened levels of air pollution may also affect decisions to work when an individual is a caretaker of dependents [17]. Most empirical evidence to date focuses on productivity [18–22] or short-run impacts of pollution shocks on hours worked [17,23]. In this paper, we use restricted individual-level panel data, paired with remote-sensing pollution estimates, to evaluate the long-term impact of chronic $PM_{2.5}$ exposure on hours worked.

Identifying the impact of chronic pollution exposure on labor supply is challenging due to data limitations and a host of confounding factors. First of all, assigning air pollution

exposure to individuals based on ground-level monitors requires strong assumptions about the spatial distribution of pollution. It is especially problematic in areas with sparse monitor coverage, which is often the case in developing countries. In addition, in China there is evidence that the official reporting of pollution data may be subject to manipulation [24]. We circumvent many of the measurement issues in individual pollution exposure by using satellite-derived pollution estimates. This is a particular advantage in rural areas, where monitoring coverage is sparse. We aggregate the pollution estimates to the sub-district level, the smallest census unit in China (i.e., villages in rural areas and neighborhoods in urban areas). We then link it to individuals in the China Family Panel Survey (CFPS), a nationwide longitudinal survey using restricted-access residences of respondents. Second, the effect of air pollution on hours worked is difficult to disentangle due to confounding factors, such as heterogeneous responses to pollution, macroeconomic shocks, and seasonality, that affect pollution exposure and labor hours simultaneously. In this study, we rely on the panel structure of the data to focus on within-individual variation in pollution. We flexibly control for common contemporaneous shocks, such as macroeconomic conditions, sector-specific trends, and seasonality of pollution. We then leverage the city-level roll-out of new monitoring requirements following Barwick et al. [25], combined with survey information on health and environmental awareness, to understand the underlying mechansims.

We find a large impact of $PM_{2.5}$ concentration on hours worked. Our individual-level fixed effects estimates suggest that a 1 $\mu g/m^3$ increase in $PM_{2.5}$ reduces the average individual's hours worked by about 14 min per week. We find that the impact is the largest for individuals with poor health, where a one-unit increase in $PM_{2.5}$ is associated with a 19 min decrease in weekly hours worked. We also find that environmental awareness significantly increases the labor supply responses to air pollution across all health statuses, with the largest increase occurring for individuals with poor health. We estimate that environmental awareness helps reduce the loss in labor supply due to sickness in individuals with poor health.

We also find that environmental regulations may affect labor supply as a result of raising the awareness of pollution. We leverage a city-level roll-out of the new National Ambient Air Quality Standards (NAAQS), which mandated the monitoring of $PM_{2.5}$ and resulted in a widespread reporting of $PM_{2.5}$ in China beginning in 2012. We find that these regulations had a significant impact on the responsiveness of individuals' hours worked to pollution. We show that the regulations had no significant immediate impact on pollution, wages, or unemployment, suggesting that the regulation did not affect hours worked through other channels.

Overall, we aim at providing empirical evidence on the long-term effect of $PM_{2.5}$ on hours worked in China. We contribute to the literature in several ways: first, we identify the impact of chronic exposure to air pollution (i.e., annual mean exposure, as opposed to acute shocks), which complements research that relies on shorter-run fluctuations in pollution in specific locations or occupations; second, by using Chinese data, we are looking at a population exposed to a very high annual mean $PM_{2.5}$ level, the impact of which is unknown in the literature; third, our study is not limited to a specific industry or city/region, as we are using a nationally representative sample with workers in a variety of sectors. Finally, we highlight the role of environmental awareness and information availability, by separating the impacts that operate through health and avoidance behavior separately.

We proceed by first describing the survey data and providing an overview of air pollution in China, its sources, the regulatory environment, and pollution data. In Section 3, we introduce our empirical approach. We then move to the empirical results and conclude.

## 2. Background and Data

### 2.1. Labor Supply

We use the restricted-access micro-data from the China Family Panel Studies (CFPS) in 2010, 2012 and 2014, which provide 94,660 observations from 35,955 individuals who were interviewed in at least two waves of the surveys across 1935 sub-districts. The survey

includes questions on the individual's employment, such as labor force participation, employment status and hours worked. Our main variable of interest is hours worked. For agricultural labor hours, we use the average hours worked per week when an individual is involved in agriculture work; for non-agricultural labor hours, we use the total number of hours worked across all current non-agricultural jobs. We do not include individuals who have both agricultural and non-agricultural jobs, as there is insufficient information in the survey on whether the jobs are concurrent. The panel structure of the data allows us to control for individual-specific time-invariant unobserved characteristics. The survey also provides rich information on the individuals and households. The restricted-access files allow us to assign pollution to individuals at the sub-district level, the smallest census unit in China.

### 2.2. Pollution Exposure

In urban areas in China, the main sources of $PM_{2.5}$ are electric power plants, industrial facilities, automobiles and heating, while in rural areas, $PM_{2.5}$ is primarily produced by biomass burning, agricultural dust, and windblown sources outside the region. According to official monitoring data, the annual mean $PM_{2.5}$ concentration across the 338 monitored cities was 50 $\mu g/m^3$ in 2016 [26], which is much higher than the 35 $\mu g/m^3$ standard set by the 2012 National Ambient Air Quality Standard (NAAQS) and the 10 $\mu g/m^3$ standard set by the World Health Organization [27].

Prior to 2012, there was no formal regulation of $PM_{2.5}$ in China and few ground-level monitors for $PM_{2.5}$. The 2012 NAAQS mandated the monitoring and reporting of $PM_{2.5}$ and set more stringent standards for other pollutants such as $PM_{10}$. The implementation of the new standards took a staggered approach, with the first phase implemented in 2012 and covering 66 cities including municipalities, provincial capitals, provincial level cities, major cities in the Jing-Jin-Ji region (also known as the national capital region), Yangzi River Delta, and Pearl River Delta; the second phase implemented in 2013 covered 116 additional cities; and the third phase implemented in 2014 added another 177 cities. By the end of 2014, all prefecture-level cities were included. Following Barwick et al. [25], we leverage the roll-out of monitoring as an information shock to households.

As of 2017, there were 1436 air pollution monitors across the country (refer to http://www.cnemc.cn/sssj/ for more information, accessed on 12 March 2020); however, the monitor coverage remains sparse even in densely-populated cities. During the period of our study, from 2009 to 2014, fewer pollution monitors were in place and even fewer recorded $PM_{2.5}$ levels. We therefore use satellite-derived annual mean $PM_{2.5}$ estimates developed by van Donkelaar et al. [28], which combine Aerosol Optical Depth (AOD) retrievals from the NASA MODIS, MISR, and SeaWIFS instruments with the GEOS-Chem chemical transport model, and are subsequently calibrated to regional ground-based observations of both the total and compositional mass using Geographically Weighted Regression (GWR). The calibration is conducted at the global scale and not for China exclusively. Monitoring $PM_{2.5}$ was not mandatory in China before 2012. The data consist of estimated annual mean $PM_{2.5}$ concentrations from 2009 to 2014 at the global scale with a grid cell resolution of $0.01° \times 0.01°$, which corresponds to roughly a square kilometer.

As discussed earlier, monitor-level $PM_{2.5}$ data are not available at the beginning of our study period; however, using monitor data from 2015 and 2016, we find a correlation of remote-sensing estimates and monitor-level averages for monitored sites to be about 0.7, without information of the composition of air-bone particulates. The mean pollution level across monitored locations is 53.4 and 49.02 $\mu g/m^3$ for 2015 and 2016, respectively, while the mean of the remote-sensing estimates is slightly lower, at 52.7 and 46.9 $\mu g/m^3$, respectively. This slight difference is expected as remote-sensing estimates tend to understate pollution at higher levels due to saturation [29]. In addition to better temporal coverage, remote-sensing pollution estimates offer several advantages over monitor-based readings for our setting, as the spatial coverage of surveyed households would be incomplete for the monitoring data. We illustrate this in Figure 1, which presents the monitor locations and heat maps of

satellite-derived pollution estimates (Figure 1a–c), and the distributions of pollution levels (Figure 1d) for Beijing, Chongqing and Shanghai, three of the largest cities in China. In all three cities, we observe sparse monitor coverage and a large within-city variation of pollution concentrations in both monitored and unmonitored areas.

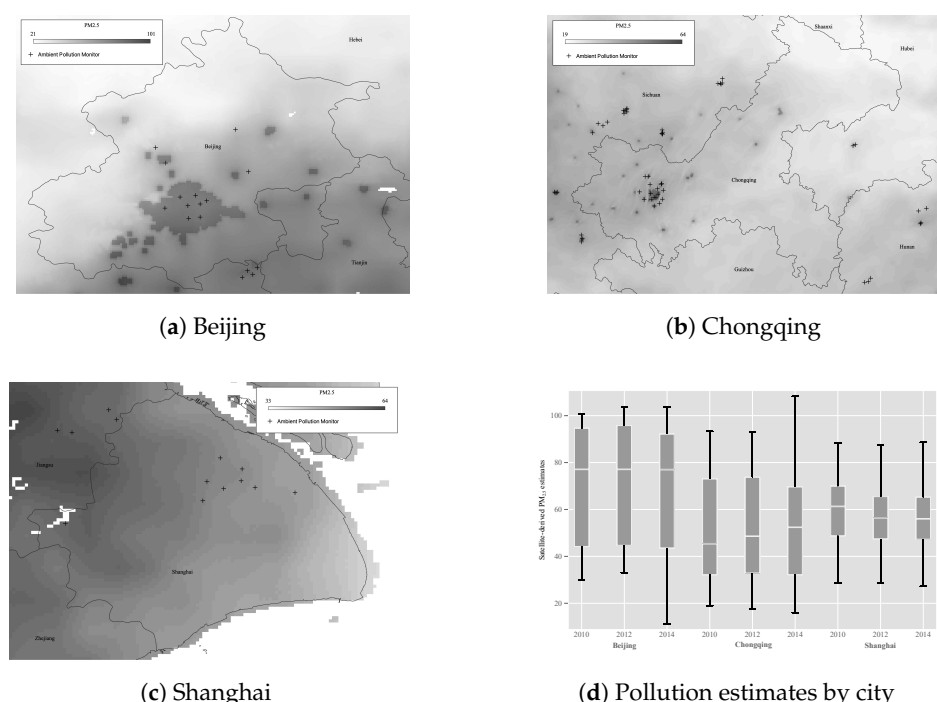

(**a**) Beijing

(**b**) Chongqing

(**c**) Shanghai

(**d**) Pollution estimates by city

**Figure 1.** Monitor Locations and Annual Mean PM$_{2.5}$ for Three Cities. Notes: The maps show the locations of ambient air pollution monitors and the satellite-derived mean PM$_{2.5}$ estimates in 2014 across Beijing (sub-figure (**a**)), Chongqing (sub-figure (**b**)) and Shanghai (sub-figure (**c**)). The three cities, Beijing, Chongqing and Shanghai, have an area of 16,808, 6340, and 82,400 km$^2$, respectively, with a population of 21.54 million, 26.32 million and 30 million. All three maps follow a northern direction on the top. Sub-figure (**d**) presents the minimum, the 10th percentile, the median, the 90th percentile and the maximum of satellite-derived pollution estimates in each city in 2010, 2012 and 2014.

We aggregate the pollution estimates for 21,592,032 grid cells within the administrative boundaries of China to the sub-district level. The number of sub-districts in a city is much larger than the number of pollution monitors. Using the same example of the cities Beijing, Chongqing, and Shanghai, the number of monitors are 12, 17 and 9, respectively; in contrast, there are 325, 1071, and 230 sub-districts in the respective cities according to the 2010 Township Population Census.

We match air pollution to individuals at the sub-district level, using the survey year and month. Table A1 in the Appendix A presents the survey schedule and the number of individuals surveyed in each month. As our data on labor hours is based on the year prior to the interview, we construct the pollution exposure measure for the same time period by calculating the weighted average of the pollution 12 months prior to the interview. For an individual living in sub-district $j$ interviewed in year $t$ and month $m$, the pollution level assigned is

$$Pollution_{jtm} = Pollution_{jt} * m/12 + Pollution_{j(t-1)} * (12-m)/12.$$

We assign pollution based on the sub-district of the individual's residential address. It is, of course, possible that individuals may be exposed to a different pollution level at the workplace. We do not observe the precise location of the workplace for respondents, but the survey indicates whether the individual works outside their home sub-district. We do

not expect, ex ante, that this measurement error would lead to any bias, but as a robustness check we also consider the sub-sample of individuals who work in the same sub-district as they reside.

### 2.3. Descriptive Statistics

About 64% of the 94,660 observations report labor force participation status, out of which 37,750 (62%) are in the labor force. Among those in the labor force, 5.7% are unemployed. The actual sample size for extensive margin models is smaller as individuals who report labor force participation and unemployment status in only one wave are dropped in individual-fixed effects estimations. Sample restrictions on missing values in key independent variables and post-migration observations also apply. The resulting sample size is 56,064 for the labor force participation model and 29,796 for the unemployment model. While 35,595 observations report hours worked, 29,389 are from individuals who are observed at least twice in the sample. We further restrict the sample by removing 3.7% of the observations with missing information on key variables, such as month of interview and sub-district identifier. We do not include the post-migration observations (about 5% of the remaining sample), as the survey does not include the timing of the move, making it impossible to accurately assign pollution exposure. The resulting sample size for our baseline specification is 25,665 observations for 11,474 individuals.

The average hours worked per week in our sample is 42.5, and those in non-agricultural sectors work almost 20 h longer than those employed only in agriculture. The within-person change in hours worked is centered around $-0.82$, with a standard deviation of 23.85. The average annual $PM_{2.5}$ for the sample is 44 $\mu g/m^3$. Non-agricultural workers face higher pollution with a $PM_{2.5}$ level of 49 $\mu g/m^3$ compared to 42 $\mu g/m^3$ for those who work in agriculture, reflecting the urban–rural pollution gap. The within-person change in $PM_{2.5}$ exposure has a mean of $-0.45$ and a standard deviation of 5.12. The distribution of hours worked and $PM_{2.5}$ (both levels and within-individual changes) are presented in Figure A1.

The baseline sample includes an equal share of males and females; about 56% of the sample has only a primary school or lower level of education; about 9% of the sample is single, while 17% co-live with children aged 7 and below. Table A2 provides the descriptive statistics for the key variables used in the analysis.

## 3. Empirical Strategy

To estimate the effect of $PM_{2.5}$ exposure on an individual's hours worked, we use a fixed effects model to isolate the within-person changes in exposure to air pollution. By controlling for individual fixed effects, we hold constant time-invariant individual characteristics. We then account for unobserved factors that affect both air pollution and labor hours, such as macroeconomic conditions, regional policies, and climate and seasonality, by including province-by-year fixed effects and interview-month fixed effects.

Our baseline model is given by the following equation:

$$H_{ijtm} = \beta Pollution_{jtm} + \gamma X_{it} + \alpha_i + \lambda_m + \delta_{pt} + \epsilon_{ijtm}, \tag{1}$$

where $H_{ijtm}$ is the average hours worked per week for an individual $i$ in sub-district $j$ during the 12-month period before year $t$ month $m$; $Pollution_{jtm}$ is the weighted annual mean $PM_{2.5}$ in sub-district $j$ during the 12-month period before year $t$ month $m$; $X_{it}$ represents time-varying individual characteristics, such as age, age-squared, education, marital status, dependent status, and employment variables such as number of jobs and sector(s); $\alpha_i$ represents time-invariant individual characteristics; $\lambda_m$ represents interview-month fixed effects; $\delta_{pt}$ represents the province by province-by-year fixed effects; and $\epsilon_{ijtm}$ represents the idiosyncratic error term.

## 4. Results

### 4.1. Extensive Margin

Before proceeding to the effect of $PM_{2.5}$ on hours worked, we first evaluate its effect at the extensive margin by estimating the impact of $PM_{2.5}$ on labor force participation and unemployment using a linear probability model. We estimate an analogue of Equation (1) with indicator variables for the status of labor force participation and unemployment as the respective dependent variables. Table 1 Column (1) shows that the effects of $PM_{2.5}$ on labor force participation (Panel A) and unemployment (Panel B) are small and statistically insignificant. Despite the different underlining labor demand and pollution conditions, this insignificant effect is consistent across urban and rural areas (Column (5)). The result is also robust across alternative model specifications (Columns (2)–(4)).

**Table 1.** Extensive Margin Impact of $PM_{2.5}$.

|  | (1) | (2) | (3) | (4) | (5) |
|---|---|---|---|---|---|
| Panel A: Labor force participation (N = 56,064) |  |  |  |  |  |
| $PM_{2.5}$ | 0.002 | 0.003 | 0.002 | 0.002 |  |
|  | (0.003) | (0.002) | (0.002) | (0.003) |  |
| $PM_{2.5} * $ Urban |  |  |  |  | 0.002 |
|  |  |  |  |  | (0.003) |
| $PM_{2.5} * $ Rural |  |  |  |  | 0.003 |
|  |  |  |  |  | (0.003) |
| Panel B: Unemployment (N = 29,796) |  |  |  |  |  |
| $PM_{2.5}$ | 0.004 | 0.002 | −0.002 | 0.004 |  |
|  | (0.004) | (0.002) | (0.004) | (0.004) |  |
| $PM_{2.5} * $ Urban |  |  |  |  | 0.004 |
|  |  |  |  |  | (0.004) |
| $PM_{2.5} * $ Rural |  |  |  |  | 0.004 |
|  |  |  |  |  | (0.004) |
| Individual characteristics | Yes | Yes | Yes | Yes | Yes |
| Individual fixed effects | Yes | Yes | Yes | Yes | Yes |
| Province-by-year fixed effects | Yes | No | No | Yes | Yes |
| Year fixed effects | No | Yes | No | No | No |
| City-by-year fixed effects | No | No | Yes | No | No |
| Month fixed effects | Yes | Yes | Yes | No | No |
| Region-by-month fixed effects | No | No | No | Yes | No |

Notes: The table presents the estimates on the impact of $PM_{2.5}$ on labor force participation (Panel A) and unemployment (Panel B). Column (1) shows the baseline results (Equation (1)) that takes into account time-varying individual characteristics, individual fixed effects, province-by-year fixed effects and month fixed effects. Urban and rural are indicator variables that take the value of 1 base on the official classification of the sub-district a respondent lives in. Columns (2) and (3) control for year fixed effects and city-by-year fixed effects, respectively, instead of province-by-year fixed effects. Column (4) replaces month fixed effects with region-by-month fixed effects. The time-varying individual characteristics include as age, age squared, education, marital status, and dependent status. Column (5) estimates the impact of $PM_{2.5}$ in urban and rural areas separately. The standard errors shown in parentheses are clustered at the city level.

### 4.2. $PM_{2.5}$ and Hours Worked

We now move to the results of our baseline model. Table 2 shows the effect of $PM_{2.5}$ on weekly hours worked by estimating Equation (1). In our preferred specification (Column (1)), we find that a 1 $\mu g/m^3$ increase in $PM_{2.5}$ reduces labor supply by 0.235 h or 14 min per week. With the sample average $PM_{2.5}$ of 44 $\mu g/m^3$ and the average number of hours worked of 42 h per week in the sample, this effect corresponds to an elasticity of roughly −0.278 (see Table A3 Column (3)).

The effect is large and economically meaningful, particularly considering the rapid pollution reduction in China in recent years. Thus, it is important to interpret this result in context. In our sample, the average within-individual change in exposure to $PM_{2.5}$ is −0.45 $\mu g/m^3$, with a standard deviation of 5.12. Thus, a one standard deviation increase in $PM_{2.5}$ is associated with a decrease of 72 min worked per week, for an average individual.

The effect of $PM_{2.5}$ on hours worked is persistent across demographic characteristics (such as age category, gender, dependent status and education), industry and income level, as shown in Figure A2.

**Table 2.** Impact of $PM_{2.5}$ on Hours Worked.

|  | (1) | (2) | (3) | (4) | (5) |
|---|---|---|---|---|---|
| $PM_{2.5}$ | −0.235 ** | −0.332 *** | −0.448 ** | −0.232 ** | −0.235 ** |
|  | (0.107) | (0.096) | (0.217) | (0.111) | (0.107) |
| Individual characteristics | Yes | Yes | Yes | Yes | Yes |
| Individual fixed effects | Yes | Yes | Yes | Yes | Yes |
| Province-by-year fixed effects | Yes | No | No | Yes | Yes |
| Year fixed effects | No | Yes | No | No | No |
| City-by-year fixed effects | No | No | Yes | No | No |
| Month fixed effects | Yes | Yes | Yes | No | Yes |
| Region-by-month fixed effects | No | No | No | Yes | No |
| Sector trend | No | No | No | No | Yes |
| N | 25,665 | 25,665 | 25,665 | 25,665 | 25,665 |

Notes: The table presents the estimates on the impact of $PM_{2.5}$ on hours worked. Column (1) shows the baseline results (Equation (1)) that takes into account time-varying individual characteristics, individual fixed effects, province-by-year fixed effects and month fixed effects. Columns (2) and (3) control for year fixed effects and city-by-year fixed effects, respectively, instead of province-by-year fixed effects. Column (4) replaces month fixed effects with region-by-month fixed effects. Column (5) includes additional controls on sector-specific time trends. The time-varying individual characteristics include age, age squared, education, marital status, dependent status, number of jobs and job sector. The standard errors shown in parentheses are clustered at city level. Statistical significance is denoted by ** for $p < 0.05$ and *** for $p < 0.01$.

*4.3. Robustness Checks*

In this section, we discuss threats to identification and tests of the robustness of our main specification.

First, in our baseline estimation, we use province-by-year fixed effects to account for macroeconomic conditions that may affect both pollution and labor supply. However, there might be concern that sub-province labor market shocks could also bias our estimates. To address this concern, we conduct a robustness check by alternatively controlling for city-by-year fixed effects and observe an even larger effect of air pollution on labor hours (Table 2 Column (3)). Our estimates are robust across other alternative specifications, such as including year fixed effects in place of province-by-year fixed effects; controlling for seasonality using region-by-month fixed effects instead of month fixed effects; and including sector-specific time trends (Table 2 Columns (2) to (5)). The magnitudes and significance of the results are also consistent across models with alternative functional forms (Table A3).

Second, there may be a concern that some individuals may have limited ability to change hours worked, which could bias our estimates. To address this possibility, we remove individuals who reported exactly 40-h working weeks in all survey waves from the baseline sample, as these individuals (206 in total) may have rigid work schedules and are thus unable to respond to heightened pollution by decreasing their hours worked. As shown in Table A4 Column (1), the effect size remains similar. We also remove the individuals with extreme hours worked and extreme changes in hours worked across waves (top and bottom 1%), as such hours, and changes may be driven by reporting errors or idiosyncratic factors other than changes in pollution. As shown in Table A4 Column (2) and (3), we find the effects to be slightly smaller but still statistically significant.

Third, assigning pollution exposure based on the residential address's sub-district could lead to a potential measurement error if pollution levels at home and at the work place are systematically different. To address this concern, we restrict our sample to individuals who live and work in the same sub-district (see Table A4 Column (3)). There may also be concerns about endogeneity, as individuals with long commutes could experience heightened pollution exposure due to traffic and fewer hours available for work. If this

were the case, we would be overestimating the effect of air pollution on labor supply. As shown in Table A4 Column (3), the estimated effect size increases slightly when we restrict the sample to individuals who live and work in the same sub-district.

Forth, our measure of hours worked is self-reported and backward-looking, so there may be concerns regarding recall bias. Because hours worked is the dependent variable of this study, as long as the measurement error due to recall bias is not correlated with other explanatory variables, our estimates will be unbiased. Nonetheless, to address this concern, we first restrict the sample to individuals who were interviewed in the summer months (May to September), when the majority of the interviews were conducted. This is because seasonal variation in the number of working hours may affect the accuracy of backward-looking survey questions. As shown in Figure A3, the reported hours worked differ substantially across the months of interviewing. We also limit the sample to individuals with better cognitive abilities, defined as those who scored higher than the median in a word-recall test administered with the survey. Neither sample restriction substantially affects the results, as shown in Table A4 Columns (4) and (5).

Last but not least, residential sorting due to pollution may bias our results, as the literature has shown that air pollution could affect long-term migration decisions in China (Chen et al. [30]). In our baseline sample, to avoid inaccurate pollution assignment, we exclude post-migration observations. However, this may lead to attrition bias. We therefore conduct a formal attrition bias test and find that moving in the next wave does not significantly affect the hours worked (Table A4 Column (6)). To understand whether there is any pollution-based sorting, we compare the characteristics of all 3566 movers in the full sample. We find no statistically significant differences in demographic characteristics nor environmental awareness between those who moved to locations with higher and lower pollution levels (Table A5). In addition, we compare the mean $PM_{2.5}$ by individual characteristics for the baseline observations (Table A6) to check for any systematic sorting before the start of the sample period and find no evidence for concern.

## 5. Heterogeneity and Mechanisms

We have established a link between $PM_{2.5}$ exposure and an individual's hours worked. In this section, we leverage a regulatory change and additional survey information to test the potential mechanisms that drive this relationship. To directly compare the effect of $PM_{2.5}$ for various sub-groups, we estimate the following equation:

$$H_{ijtm} = \Sigma_{s=1}^{N} \beta_s Pollution_{jtm} \times S_{it} + \gamma X_{it} + \alpha_i + \lambda_m + \delta_{pt} + \epsilon_{ijtm} \qquad (2)$$

where $S_{it}$ represents the sub-group indicators and $\beta_s$ measures the effect of $PM_{2.5}$ on hours worked for sub-group $S$.

First, we expect the effect of $PM_{2.5}$ on hours worked to differ by health status, as individuals with poor health are more likely to be impacted by air pollution. To test this, we divide the baseline sample into three sub-groups by self-rated health status (self-rated health status has a 5-point scale: 1 = excellent, 2 = very good, 3 = good, 4 = fair and 5 = poor. We consider the first three as good health): good (48.9%), fair (38.7%) and poor (12.4%). As self-rated health may change across surveys, we also include it as an additional control along with other time-varying individual characteristics. As shown in Table 3 Column (1), we find that individuals with poor health respond the most to pollution changes: a one-unit increase in $PM_{2.5}$ reduces labor supply by 0.319 h or 19 min per week. In comparison, the responses by individuals with fair and good health are roughly 4 and 6 min smaller (0.065 and 0.107 h per week, respectively).

Second, the effect of $PM_{2.5}$ on labor hours is possibly nonlinear in $PM_{2.5}$. When pollution levels are low, the impact of a marginal change in $PM_{2.5}$ on health and avoidance behavior may be different from that at high levels. To test this, we allow for differential responses for high- and low-pollution areas. As shown in Table 3 Column (2), we find the response to changes in air pollution to be small and statistically insignificant when the $PM_{2.5}$ level is low, defined as below the 35 $\mu g/m^3$ standards set by 2012 NAAQS (31% of

the sample). Our estimates also show that, perhaps unsurprisingly, when the pollution level is low, there is a minimal labor hour response across all health statuses (Table 4 Panel A Column (1)). Sub-group sample distributions by pollution level and health status are presented in Figure A4.

Third, the labor supply response to air pollution through avoidance behavior may be linked to how aware one is about pollution levels. To test this, we rely on respondents' average awareness of environmental issues in the 2012 and 2014 surveys as this question was not available in 2010. In our sample, about 16% of the respondents state that they are unaware of any environmental issues in the country (i.e., "uninformed" individuals). We find that the labor supply response to changes in $PM_{2.5}$ is small and statistically insignificant for this group of individuals (Table 3 Column (2)). On the contrary, weekly hours worked decreases by 0.26 h, i.e., 16 min, in response to a one-unit increase in $PM_{2.5}$ for individuals who claim to be environmentally aware (i.e., "informed" individuals). Our results also show that the labor hour responses are only statistically significant for informed individuals when the pollution exposure is higher than the official standards, as shown in Table 4 Panel B Columns (3) and (4).

**Table 3.** Heterogeneous Effect of $PM_{2.5}$ on Hours Worked: Self-rated Health, Awareness and Pollution Level.

| | (1) | (2) | (3) | (4) |
|---|---|---|---|---|
| Good ($\beta_1$) | −0.212 ** | | | |
| | (0.100) | | | |
| Fair ($\beta_2$) | −0.254 ** | | | |
| | (0.102) | | | |
| Poor ($\beta_3$) | −0.319 *** | | | |
| | (0.109) | | | |
| Low pollution ($\lambda_0$) | | 0.026 | | |
| | | (0.200) | | |
| High pollution ($\lambda_1$) | | −0.261 ** | | |
| | | (0.106) | | |
| Unaware ($\theta_0$) | | | −0.155 | |
| | | | (0.147) | |
| Aware ($\theta_1$) | | | −0.260 ** | |
| | | | (0.102) | |
| Pre-NAAQS ($\eta_0$) | | | | −0.183 * |
| | | | | (0.103) |
| After-NAAQS ($\eta_1$) | | | | −0.279 *** |
| | | | | (0.106) |
| $\beta_2 - \beta_1$ | −0.042 ** | | | |
| | (0.020) | | | |
| $\beta_3 - \beta_1$ | −0.107 *** | | | |
| | (0.037) | | | |
| $\beta_3 - \beta_2$ | −0.065 * | | | |
| | (0.034) | | | |
| $\lambda_1 - \lambda_0$ | | −0.287 ** | | |
| | | (0.142) | | |
| $\theta_1 - \theta_0$ | | | −0.104 | |
| | | | (0.096) | |
| $\eta_1 - \eta_0$ | | | | −0.096 *** |
| | | | | (0.032) |
| Individual characteristics | Yes | Yes | Yes | Yes |
| Individual fixed effects | Yes | Yes | Yes | Yes |
| Province-by-year fixed effects | Yes | Yes | Yes | Yes |
| Month fixed effects | Yes | Yes | Yes | Yes |
| N | 25,665 | 25,665 | 25,665 | 25,665 |

Notes: The table presents the estimates on the heterogeneous effects of $PM_{2.5}$ on hours worked by self-rated health (Column (1)), pollution level Columns (2)), awareness of environmental issues (Column (3)), and the implementation of NAAQS (Columns (4)) by estimating Equation (2). *Good* is an indicator variable that takes the value of 1 if a respondent has a self-rate health status of excellent, very good or good. *Fair* and *Poor* are indicator variables that take the value of 1 if the respondent has a self-rate health status of fair and poor, respectively. *Low* and *High* are indicator variables that take the value of 1 if the average pollution exposure of a response is below and above the sample mean, respectively. All models account for time-varying individual characteristics, individual fixed effects, province-by-year fixed effects and month fixed effects. The time-varying individual characteristics include age, age squared, education, marital status, dependent status, number of jobs and job sector. We also control for the self-rated health and pollution category, respectively, for Columns (1) and (2). The standard errors shown in parentheses are clustered at the city level. Statistical significance is denoted by * for $p < 0.1$, ** for $p < 0.05$ and *** for $p < 0.01$.

**Table 4.** Heterogeneous Effect on Hours Worked: Interaction between Health, Awareness and Pollution Level.

| | (1) | (2) Pollution Level | (3) | (4) | (5) Awareness | (6) |
|---|---|---|---|---|---|---|
| | Low | High | Diff | Unaware | Aware | Diff |
| **Panel A: By health status** | | | | | | |
| Good ($\beta_1$) | 0.090 | −0.234 *** | −0.324 ** | −0.128 | −0.236 *** | −0.232 * |
| | (0.145) | (0.069) | (0.144) | (0.106) | (0.068) | (0.126) |
| Fair ($\beta_2$) | 0.037 | −0.282 *** | −0.319 ** | −0.188 * | −0.274 *** | −0.250 * |
| | (0.147) | (0.069) | (0.147) | (0.111) | (0.069) | (0.132) |
| Poor ($\beta_3$) | −0.074 | −0.364 *** | −0.291 * | −0.215 *** | −0.345 *** | −0.289 ** |
| | (0.159) | (0.077) | (0.152) | (0.118) | (0.074) | (0.138) |
| $\beta_2 - \beta_1$ | 0.022 | −0.048 ** | 0.005 | −0.059 ** | −0.038 ** | −0.018 |
| | (0.052) | (0.024) | (0.029) | (0.024) | (0.019) | (0.021) |
| $\beta_3 - \beta_1$ | −0.109 | −0.013 *** | 0.033 | −0.086 ** | −0.109 *** | −0.057 * |
| | (0.088) | (0.042) | (0.046) | (0.039) | (0.034) | (0.032) |
| $\beta_3 - \beta_2$ | −0.131 | −0.083 ** | 0.028 | −0.027 | −0.071 ** | −0.038 |
| | (0.083) | (0.039) | (0.046) | (0.036) | (0.030) | (0.027) |
| **Panel B: Pollution level** | | | | | | |
| Low ($\lambda_0$) | | | | 0.122 | −0.014 | −0.136 |
| | | | | (0.186) | (0.144) | (0.132) |
| High ($\lambda_1$) | | | | −0.110 | −0.292 *** | −0.182 * |
| | | | | (0.111) | (0.070) | (0.105) |
| $\lambda_1 - \lambda_0$ | | | | −0.232 | −0.278 ** | 0.046 |
| | | | | (0.157) | (0.142) | (0.068) |
| Individual characteristics | | Yes | | | Yes | |
| Individual fixed effects | | Yes | | | Yes | |
| Province-by-year fixed effects | | Yes | | | Yes | |
| Month fixed effects | | Yes | | | Yes | |
| N | | 25,665 | | | 25,665 | |

Notes: The table presents the estimates on the heterogeneous effects of $PM_{2.5}$ on hours worked by the interaction of self-rated health, pollution level and awareness of environmental issues by estimating Equation (2). *Good* is an indicator variable that takes the value of 1 if a respondent has a self-rate health status of excellent, very good or good. *Fair* and *Poor* are indicator variables that take the value of 1 if the respondent has a self-rate health status of fair and poor, respectively. *Low* and *High* are indicator variables that take the value of 1 if the average pollution exposure of a respond is below and above the sample mean, respectively. All models account for time-varying individual characteristics, individual fixed effects, province-by-year fixed effects, and month fixed effects. The time-varying individual characteristics include age, age squared, education, marital status, dependent status, number of jobs, job sector, health status and pollution level. The standard errors shown in parentheses are clustered at the city level. Statistical significance is denoted by * for $p < 0.1$, ** for $p < 0.05$ and *** for $p < 0.01$.

We note that environmental awareness could be correlated with health, as individuals with health concerns such as respiratory and cardiovascular diseases may pay more attention to air pollution in order to avoid excess exposure and the resulting adverse health effects [31]. To address this concern, we further interact environmental awareness with self-rated health and find that, without environmental awareness, the labor supply of individuals with good health tends to be unresponsive to $PM_{2.5}$ (Table 4 Panel A Column (4)). Weekly hours worked by individuals with poor health, on the other hand, decreases by 0.215 h (13 min), and this response is significantly different from that of individuals with good health. As uninformed individuals are less likely to take precautions against air pollution, the effect we find for this group could be viewed as the health effect. In contrast, across all health statuses, informed individuals respond more to air pollution (Table 4 Panel A Column (6)). For healthy individuals, this response could be interpreted mostly as avoidance behavior; however, for individuals with poorer health, this would be a combined effect operating through health and avoidance behavior. We find the change in labor supply, driven by environmental awareness, to be significantly larger ($\beta_3 - \beta_1$ in Table 4 Column (6)) for individuals with poor health relative to those with good health, which suggests that unhealthy individuals are more proactive in engaging in avoidance behavior. Although we are unable to directly separate the effect through health and avoidance behavior, we infer that environmental awareness reduces the lost hours worked due to sickness by at least 6 min per week. The combined health and avoidance behavior effect for individuals with poor health is a reduction of labor supply by 0.345 h. As the effect of avoidance behavior on the individual with poor health is at least 0.236 h ($\beta_1$ in Table 4 Column (5)), for the estimated effect for individuals with good health, the health effect is at the most 0.109 h, which is 6 min or 0.106 h less than the effect on individuals with poor health that are unaware of air pollution (0.215 or $\beta_3$ in Table 4 Column (4)).

*Awareness and New National Ambient Air Quality Standards*

As environmental awareness increases the responsiveness to pollution, changes in environmental awareness also plays a role in individuals' labor supply decisions. Barwick et al. [25] show that real-time air quality monitoring and the accompanying disclosure program, brought about by the staggered implementation of the 2012 NAAQS, substantially increased households' awareness about ambient air pollution. The last survey wave we used was conducted in 2014, before the full roll-out of the new standards, but we still find significant increases in environmental awareness for individuals residing in the cities where the new regulation was implemented (Table A7 Panel B). Using post-NAAQS as an indicator for higher pollution awareness, we also find that the labor hour responses increase by almost 6 min (0.096 h), as shown in Table 3 Column (3).

For our interpretation to be valid, the 2012 NAAQS should not affect air pollution or economic behavior directly. This can be tested directly, and we find that the regulation did not lead to a reduction in pollution during our sample period (Table A7 Panel A), which is consistent with findings in Barwick et al. [25]. There were likely eventually impacts associated with the implementation of these standards, as local governments subsequently began regulating pollution sources, but to the extent that is true, it happened after our period of study. We also estimate the effect of the 2012 NAAQS on labor force participation, unemployment and wages (Table A8) and find no credible evidence of any impact. This does not contradict the exiting literature. In a study on the 1990 Clean Air Act Amendments in the United States, Sheriff et al. [32] find that the employment effect became significant three years after its detailed implementation. Our findings therefore suggest that the larger labor supply responses to $PM_{2.5}$ upon the implementation of the 2012 NAAQS is likely a result of the increasing awareness of air pollution due to the information campaign accompanying the introduction of pollution monitoring.

## 6. Conclusions

Despite widespread concerns about ambient air pollution, relatively little is known about the impact of long-run exposure on an individual's labor supply decisions. We contribute to this line of empirical literature by evaluating the effect of $PM_{2.5}$ on labor hours in China. We use remote-sensing estimates of $PM_{2.5}$ to assign pollution exposure to individuals in the panel survey of CFPS at the sub-district level, the smallest census unit in China. We find that the impact of chronic exposure to air pollution on labor hours to be large and significant. Our individual fixed effects estimates indicate that a one-unit increase in $PM_{2.5}$ concentration is associated with a 14 min per week decrease in hours worked. In comparison, Aragón et al. [17] find that a 10 $\mu g/m^3$ reduction in $PM_{2.5}$ in the short-run is associated with an increase of 1.9 h worked (equivalent to 11.4 min per unit). The results suggest that researchers and policymakers should take into account not only productivity impacts, but also labor supply effects, when considering policies to reduce pollution levels.

Our result is closely related to Aragón et al. [17], in which the authors estimate that 1 $\mu g/m^3$ of an increase in air pollution reduces the working hours by 19 min for households with susceptible dependents.

In addition to our core finding—that individuals are responsive to long-run exposure to $PM_{2.5}$—our results also help isolate the mechanisms of the effect, which has important policy implications. We show that individuals with poor health are more responsive to $PM_{2.5}$ concentrations. For those who are uninformed about air pollution, this effect is likely through the impact of $PM_{2.5}$ on health. Individuals who are environmentally aware also reduce hours worked in response to $PM_{2.5}$, but this appears to be driven by avoidance behavior. This applies to both healthy and unhealthy individuals, but individuals with poor health are more responsive when they claim to be "environmentally aware". We further estimate that environmental awareness reduces the loss in labor supply due to sickness for individuals with poor self-rated health.

We highlight the importance of providing information about pollution to the public. In the case of China, we find that the staggered implementation of the 2012 NAAQS helped

in raising awareness about air pollution and it affected labor supply decisions through the change in awareness. We conclude by noting that this is a first step in understanding the impact of changing long-run pollution levels on labor supply decisions. More research is needed to understand the underlying mechanisms, equilibrium impacts and welfare effects.

**Author Contributions:** Both authors (M.F. and C.G.) contributed to the research design, implementation, data analysis and writing. All authors have read and agreed to the published version of the manuscript.

**Funding:** Grainger acknowledges financial support from the William F. Vilas Trust Estate (Early Career Award AAA3364-5).

**Data Availability Statement:** Not applicable.

**Acknowledgments:** For comments and suggestions on earlier drafts, we thank Ian Bateman, Antonio Bento, Ian Coxhead, Olivier Deschenes, Josh Graff-Zivin, Michael Greenstone, Rema Hanna, Paulina Oliva, Dominic Parker, Daniel Phaneuf, and seminar participants at the Norwegian School of Economics, National University of Singapore, University of California Santa Barbara, University of Exeter, University of Southern California, University of Wisconsin-Madison, World Congress for Environmental and Resource Economics and IZA Workshop on Environment and Labor Markets. We are particularly grateful to Yang Yao for early discussions.

**Conflicts of Interest:** The authors declare no conflict of interest.

## Appendix A

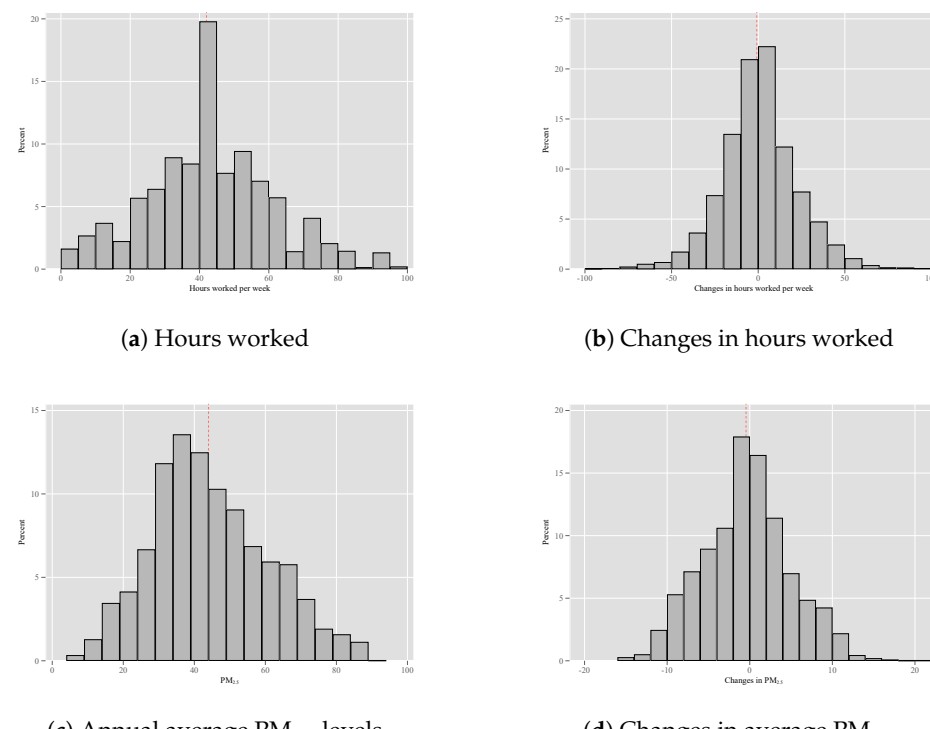

(**a**) Hours worked

(**b**) Changes in hours worked

(**c**) Annual average PM$_{2.5}$ levels

(**d**) Changes in average PM$_{2.5}$

**Figure A1.** Hours Worked and Particulate Concentrations (Levels and Changes). Notes: The figures show the distribution of hours worked (sub-figure (**a**)), changes in hours worked (sub-figure (**b**)), PM$_{2.5}$ (sub-figure (**c**)) and changes in PM$_{2.5}$ (sub-figure (**d**)). The red dotted lines indicate the mean hours worked per week (42 h), mean changes in hours worked ($-0.82$ h), mean exposure to PM$_{2.5}$ (42 µg/m$^3$), and mean changes in exposure ($-0.45$ µg/m$^3$), respectively.

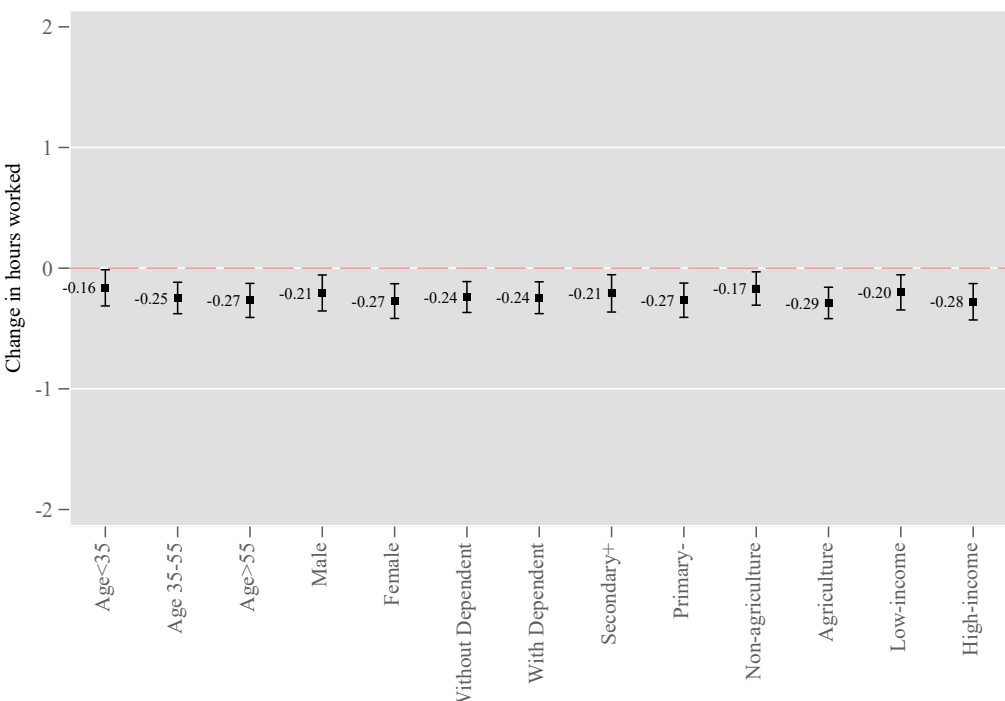

**Figure A2.** Heterogeneous Effect of PM$_{2.5}$ on Hours Worked: Demographic Characteristics, Industry and Income. Notes: The figure presents the coefficients (black square) and 95% confidence intervals (black line) for the effect of PM$_{2.5}$ on hours worked per week by estimating Equation (2). The red dot line indicates the reference value of 0.

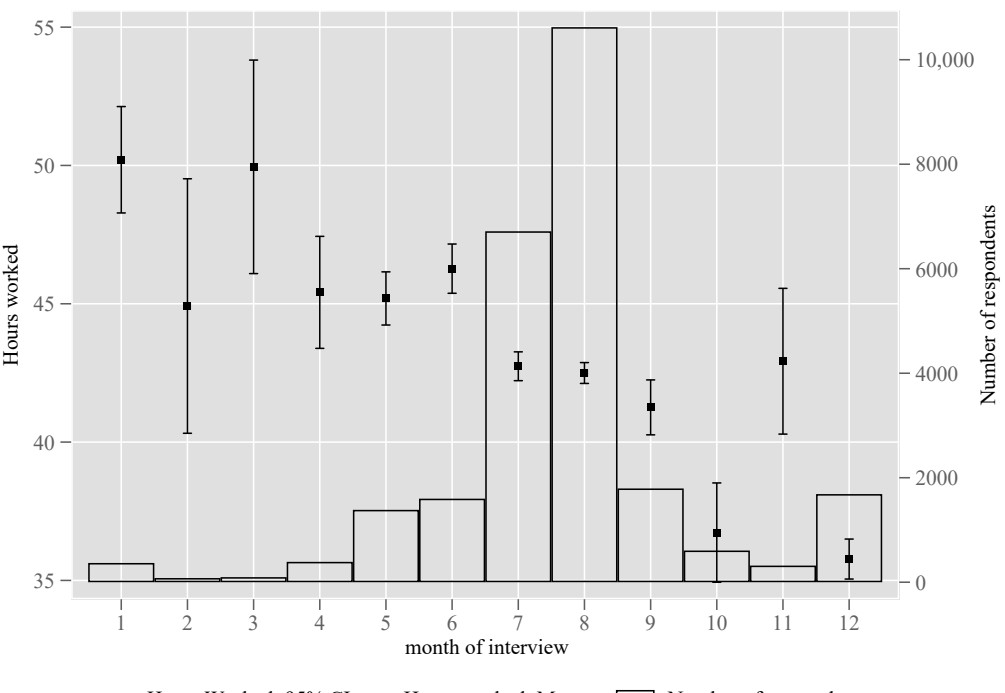

**Figure A3.** Labor Hours by Month of Interview. Notes: The graph shows the average hours worked per week and the number of respondents by interview month.

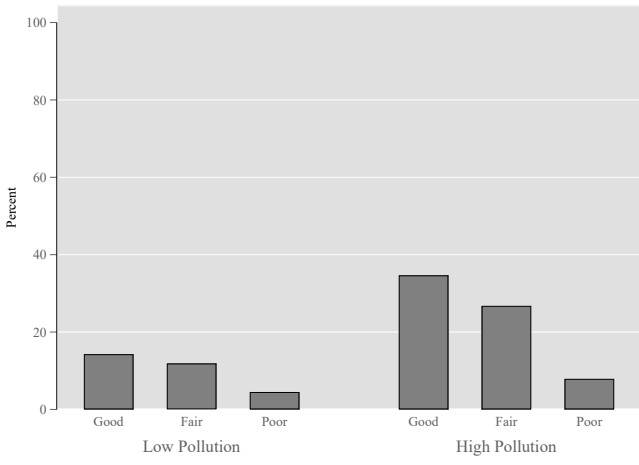

(**a**) By health status and pollution level

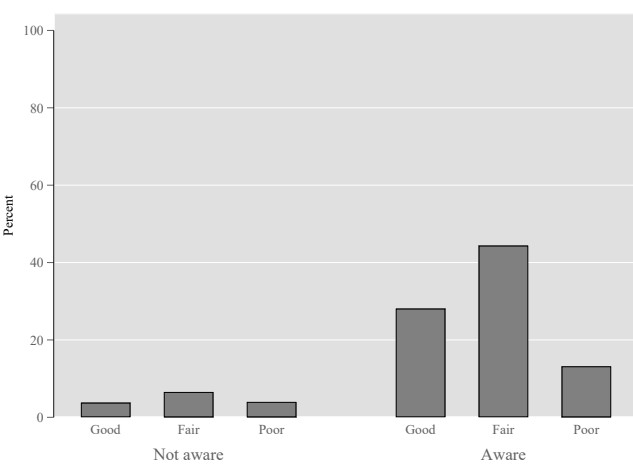

(**b**) By health status and awareness

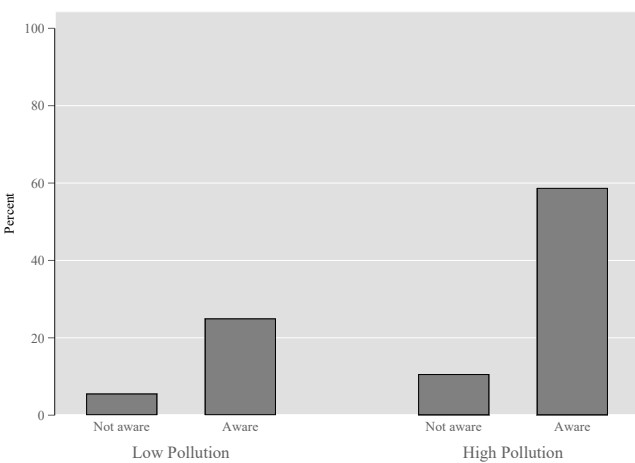

(**c**) By awareness and pollution level

**Figure A4.** Sample Distribution by Health Status, Pollution Level and Awareness. Notes: The figures show the percentage of sample in each sub-group by health status, pollution level and awareness of environmental issues.

**Table A1.** Interview Year and Month.

| Month | Year of Interview | | | | | | Total |
|---|---|---|---|---|---|---|---|
| | **2010** | **2011** | **2012** | **2013** | **2014** | **2015** | |
| Jan | 0 | 96 | 0 | 18 | 0 | 254 | 368 |
| Feb | 0 | 6 | 0 | 60 | 0 | 14 | 80 |
| Mar | 0 | 45 | 0 | 13 | 0 | 39 | 97 |
| Apr | 380 | 0 | 0 | 0 | 0 | 4 | 390 |
| May | 1367 | 0 | 0 | 0 | 0 | 18 | 1385 |
| Jun | 1596 | 0 | 0 | 0 | 0 | 3 | 1599 |
| Jul | 2417 | 0 | 1309 | 0 | 2991 | 0 | 6717 |
| Aug | 2451 | 0 | 4643 | 0 | 3530 | 0 | 10,624 |
| Sep | 90 | 7 | 1120 | 0 | 577 | 0 | 1794 |
| Oct | 29 | 0 | 43 | 0 | 534 | 0 | 606 |
| Nov | 35 | 0 | 33 | 0 | 250 | 0 | 318 |
| Dec | 115 | 0 | 1430 | 0 | 142 | 0 | 1687 |
| Total | 8486 | 154 | 8578 | 91 | 8024 | 332 | 25,665 |

Notes: The table presents the number of interviews conducted by interview year and month.

**Table A2.** Descriptive Statistics.

| | (1)<br>**All** | (2)<br>**Agriculture** | (3)<br>**Non-Agriculture** |
|---|---|---|---|
| Hours worked per week | 42.47 | 35.79 | 54.76 |
| | (20.30) | (16.07) | (21.49) |
| $PM_{2.5}$ | 44.27 | 41.94 | 48.57 |
| | (16.32) | (16.53) | (15.01) |
| Age | 46.06 | 49.37 | 39.97 |
| | (12.49) | (12.10) | (10.79) |
| Gender: male = 1 | 0.50 | 0.44 | 0.62 |
| | (0.50) | (0.50) | (0.49) |
| Education: below primary = 1 | 0.56 | 0.73 | 0.25 |
| | (0.50) | (0.44) | (0.44) |
| Marital status: single = 1 | 0.09 | 0.07 | 0.13 |
| | (0.29) | (0.26) | (0.34) |
| Dependent: yes = 1 | 0.17 | 0.13 | 0.24 |
| | (0.37) | (0.34) | (0.43) |
| Observations | 25,665 | 16,627 | 9,038 |

Notes: The table presents the sample statistics for the key variables for the full sample (Column (1)), those who work in agriculture (Column (2)) and those who work in the non-agricultural sector (Column (3)). The standard deviations are provided in parentheses and the sample size at the bottom of the panel.

**Table A3.** Impact of $PM_{2.5}$ on Hours Worked: Alternative Functional Form.

| | (1)<br>**Hours** | (2)<br>**Log Hours** | (3)<br>**Log Hours** | (4)<br>**Hours** |
|---|---|---|---|---|
| Log $PM_{2.5}$ | −7.673 ** | | −0.278 * | |
| | (3.735) | | (0.141) | |
| $PM_{2.5}$ | | −0.009 ** | | |
| | | (0.004) | | |
| Standardized $PM_{2.5}$ | | | | −3.841 ** |
| | | | | (1.639) |
| Individual characteristics | Yes | Yes | Yes | Yes |
| Individual fixed effects | Yes | Yes | Yes | Yes |
| Province-by-year fixed effects | Yes | Yes | Yes | Yes |

**Table A3.** *Cont.*

|  | (1)<br>Hours | (2)<br>Log Hours | (3)<br>Log Hours | (4)<br>Hours |
|---|---|---|---|---|
| Month fixed effects |  |  |  |  |
| N | 25,665 | 25,665 | 25,665 | 25,665 |

Notes: The table presents the estimates on the impact of $PM_{2.5}$ on hours worked using alternative functional forms. Column (1) uses the log-linear functional form and estimates the effect of 1% increase in $PM_{2.5}$ on hours worked. Column (2) uses the linear-log specification and estimates the effect of 1 $\mu g/m^3$ increase of $PM_{2.5}$ on the weekly hours worked in percentage terms. Column (3) uses the log-log specification and estimates the effect of the 1% increase in $PM_{2.5}$ on the hours worked in percentage terms. Column (4) uses standardized $PM_{2.5}$ and estimates the effect of one standard deviation increase in $PM_{2.5}$ on hours worked. All the models include time-varying individual characteristics, individual fixed effects, province-by-year fixed effects and month fixed effects. The time-varying individual characteristics include age, age squared, education, marital status, dependent status, number of jobs and job sector. The standard errors shown in parentheses are clustered at city level. Statistical significance is denoted by * for $p < 0.1$ and ** for $p < 0.05$.

**Table A4.** Impact of $PM_{2.5}$ on Hours Worked: Robustness Checks.

|  | (1)<br>Exclude<br>40 h | (2)<br>Extreme<br>Hours | (3)<br>Extreme<br>Changes in Hours | (4)<br>Location | (5)<br>Interview<br>Month | (6)<br>Memory | (7)<br>Attrition |
|---|---|---|---|---|---|---|---|
| $PM_{2.5}$ | −0.242 ** | −0.201 ** | −0.208 ** | −0.280 ** | −0.205 * | −0.256 ** | −0.239 ** |
|  | (0.108) | (0.009) | (0.100) | (0.139) | (0.123) | (0.107) | (0.100) |
| Moved |  |  |  |  |  |  | 2.138 |
|  |  |  |  |  |  |  | (1.845) |
| Individual characteristics | Yes | Yes | Yes | Yes | Yes | Yes | Yes |
| Individual fixed effects | Yes | Yes | Yes | Yes | Yes | Yes | Yes |
| Province-by-year fixed effects | Yes | Yes | Yes | Yes | Yes | Yes | Yes |
| Month fixed effects | Yes | Yes | Yes | Yes | Yes | Yes | Yes |
| N | 25,148 | 24,937 | 25,062 | 17,102 | 19,798 | 23,535 | 25,665 |

Notes: The table presents the estimates for various robustness checks on the effect of $PM_{2.5}$ on hours worked. Columns (1)–(6) use restricted samples to estimate Equation (1): Column (1) excludes individuals that reported a 40-hour working week for all the surveys; Column (2) excludes observations with the top and bottom 1% in hours worked per week; Column (3) excludes observations with the top and bottom 1% of changes in hours worked per week; Column (4) includes only individuals who live and work in the same sub-district; Column (5) includes only individuals interviewed between May and September; and Column (6) includes only individuals with an above-median score for the word recall test. Column (7) presents the estimates from the attrition bias test by including the indicator variable of what moved in the next wave. All the models include time-varying individual characteristics, individual fixed effects, province-by-year fixed effects and month fixed effects. The time-varying individual characteristics include as age, age squared, education, marital status, dependent status, number of jobs and job sector. The standard errors shown in parentheses are clustered at the city level. Statistical significance is denoted by * for $p < 0.1$ and ** for $p < 0.05$.

**Table A5.** Mover Characteristics.

|  | (1)<br>Increased Pollution | (2)<br>Reduced Pollution | (3)<br>Std. Diff. |
|---|---|---|---|
| Age | 39.94 | 41.31 | −0.08 |
|  | (16.02) | (16.02) |  |
| Gender: male = 1 | 0.49 | 0.49 | 0.08 |
|  | (0.50) | (0.50) |  |
| Education: below primary = 1 | 0.38 | 0.43 | −0.09 |
|  | (0.49) | (0.50) |  |
| Marital status: single = 1 | 0.27 | 0.23 | 0.07 |
|  | (0.44) | (0.43) |  |
| Dependent: yes = 1 | 0.18 | 0.19 | −0.01 |
|  | (0.39) | (0.39) |  |
| Environmental awareness | 5.93 | 5.80 | 0.04 |
|  | (2.83) | (3.25) |  |
| Observations | 2213 | 1353 |  |

Notes: This table provides the statistics for mover characteristics. Columns (1) and (2) summarizes the characteristics for movers who moved to sub-districts with higher and lower $PM_{2.5}$, respectively. Column (3) reports standardized differences between Columns (1) and (2). The standard deviations are provided in parentheses and number of moves at the bottom of the table.

**Table A6.** Difference in Baseline PM$_{2.5}$ by Group.

| | (1) Comparison Group | (2) Reference | (3) Std. Diff. |
|---|---|---|---|
| Age < 35 (Ref = Other age) | 45.08 | 44.92 | 0.01 |
| | (17.73) | (15.89) | |
| Age 35–55 (Ref = Other age) | 44.64 | 45.34 | −0.04 |
| | (15.81) | (16.93) | |
| Age > 55 (Ref = Other age) | 45.65 | 44.78 | 0.05 |
| | (15.92) | (16.8) | |
| Male (Ref = Female) | 44.92 | 45.00 | 0.01 |
| | (16.34) | (16.40) | |
| Below primary (Ref = Above primary) | 48.15 | 42.35 | 0.36 |
| | (15.75) | (16.41) | |
| Single (Ref = Married) | 43.91 | 45.08 | 0.07 |
| | (16.75) | (16.32) | |
| Dependents (Ref = No Dependents) | 44.93 | 44.96 | −0.02 |
| | (17.35) | (16.17) | |

Notes: This table presents the mean PM$_{2.5}$ and normalized differences between individuals with varied characteristics.

**Table A7.** Effect of NAAQS on Pollution and Environmental Awareness.

| | (1) | (2) | (3) |
|---|---|---|---|
| Panel A: PM$_{2.5}$ (N = 25,655) | | | |
| NAAQS * Post | 0.154 | 0.080 | 0.165 |
| | (0.458) | (0.527) | (0.444) |
| Panel B: Envirnmental awareness (N = 10,560) | | | |
| NAAQS * Post | 0.242 * | 0.221 ** | 0.316 ** |
| | (0.125) | (0.104) | (0.127) |
| Individual characteristics | Yes | Yes | Yes |
| Individual fixed effects | Yes | Yes | Yes |
| Province-by-year fixed effects | Yes | No | No |
| Year fixed effects | No | Yes | No |
| Month fixed effects | Yes | Yes | No |
| Region-by-month fixed effects | No | No | Yes |

Notes: The table presents the estimates of the effect of NAAQS on PM$_{2.5}$ (Panel A) and environmental awareness (Panel B). Column (1) takes into account time-varying individual characteristics, individual fixed effects, province-by-year fixed effects and month fixed effects. Column (2) controls for year fixed effects instead of province-by-year fixed effects. Column (3) replaces month fixed effects with region-by-month fixed effects. The time-varying individual characteristics include age, age squared, education, marital status, dependent status, number of jobs and job sector. The standard errors shown in parentheses are clustered at the city level. Statistical significance is denoted by * for $p < 0.1$ and ** for $p < 0.05$.

**Table A8.** Effect of NAAQS on Labor Outcomes.

| | (1) | (2) | (3) |
|---|---|---|---|
| Panel A: Labor force participation (N = 56,064) | | | |
| PM$_{2.5}$ | 0.004 | 0.004 | 0.004 |
| | (0.003) | (0.003) | (0.003) |
| NAAQS * Post | 0.034 | 0.011 | 0.036 |
| | (0.056) | (0.051) | (0.054) |
| PM$_{2.5}$ * NAAQS * Post | −0.001 | −0.001 | −0.001 |
| | (0.001) | (0.001) | (0.001) |
| Panel B: Unemployment (N = 29,796) | | | |
| PM$_{2.5}$ | 0.002 | 0.002 | 0.002 |
| | (0.003) | (0.002) | (0.002) |
| NAAQS * ost | −0.039 | 0.016 | −0.036 |
| | (0.029) | (0.034) | (0.029) |
| PM$_{2.5}$ * NAAQS * Post | 0.001 | −0.000 | 0.001 |
| | (0.001) | (0.000) | (0.001) |

**Table A8.** *Cont.*

|  | (1) | (2) | (3) |
|---|---|---|---|
| Panel C: Log of wage (N = 15,564) |  |  |  |
| PM$_{2.5}$ | −0.002 | −0.002 | −0.002 |
|  | (0.004) | (0.004) | (0.004) |
| NAAQS ∗ Post | 0.014 | 0.020 | 0.020 |
|  | (0.115) | (0.101) | (0.114) |
| PM$_{2.5}$ ∗ NAAQS ∗ Post | 0.000 | 0.000 | 0.000 |
|  | (0.002) | (0.002) | (0.002) |
| Individual characteristics | Yes | Yes | Yes |
| Individual fixed effects | Yes | Yes | Yes |
| Province-by-year fixed effects | Yes | No | Yes |
| Year fixed effects | No | Yes | No |
| Month fixed effects | Yes | Yes | No |
| Region-by-month fixed effects | No | No | Yes |

Notes: The table presents the estimates of the effect of NAAQS on labor force participation (Panel A), unemployment (Panel B) and wage (Panel C). Column (1) takes into account time-varying individual characteristics, individual fixed effects, province-by-year fixed effects and month fixed effects. Column (2) controls for year fixed effects instead of province-by-year fixed effects. Column (3) replaces month fixed effects with region-by-month fixed effects. The time-varying individual characteristics include age, age squared, education, marital status, dependent status, number of jobs and job sector. The standard errors shown in parentheses are clustered at the city level.

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
