# Peer review of "The Impact of Air Pollution on Labor Supply in China"

_sustainability, doi:10.3390/su151713082_

Round 1

Reviewer 1 Report

in the presented work, Impact of Air Pollution on Labor Supply in China is presented with good discussions and clear results.
this type of papers are sounded and improved the scientific view of the society. Also, this work within the journal scope  and can accept after minor revision

English needs revision use scientific passive language no need to write WE… in each paragraph for each verb. the work aims should be stated clear at the end of introduction with the clear work novelty 

Needs revision 

Author Response

Thank you very much for your comments. We have updated the introduction to state the aims and contribution of the paper. We would, however, leave the use of passive language to copyedit stage.

Reviewer 2 Report

I would say this is very important research, that the findings have important implications to support regulations, and provide a basis for economic evaluation of productivity loss, in the context of rulemaking.  I do think it would be good to hear from the authors why more recent data was not considered, and the cutoff was so long ago. 

I find two simple comments. It would be good to possibly place a table to compare these results with similar research in other countries. I would also make the graphs in color, for clarity. 

Author Response

Thank you very much for your comments. Please find our responses below:

  1. Although the survey data (CPFS) is available until 2020, the publicly accessible data does not have location identifier and does not allow matching of pollution exposure to individual location. The matching using restricted sub-districted ID was only provided for the study period.
  2. We highlighted the comparison to results in other similar research following your comments.
  3. We hope to see you understanding that the graphs are made with color-blind friendly scheme, and we prefer to keep it this way.

Reviewer 3 Report

Keywords

-       Ensure keywords does not repeat part of the manuscript title.

Background and Data

-       Labor Supply

o   Provide the ethical considerations.

-       Pollution Exposure

o   Does China have new ambient air quality standard for PM2.5?

Results

-       Page 7, line 316-319, provide the reference.

Discussion

-       Provide suggestions for future research.

Conclusion

-       Summarize the manuscript without being redundant and express the impact of the study.

Figure 1

-       Provide the direction and scale of maps.

Table 1

-       Provide the definition of urban and rural.

Table 3 and Table 4

-       Provide the definition of each self-rated health, low pollution, and high pollution.

Author Response

Thank you very much for your comments. Our explanations and revisions are as follows:

  1. The keywords have been updated accordingly.
  2. On background the data:
    • This paper uses secondary data without any interaction with human subjects. The data used does not include identifiable private information. Following the requirement by IRB of University of Wisconsin-Madison, it falls under the broad category of " Analysis of publicly available dataset", with a responsible use agreement signed with the data provider and does not require an IRB.
    • Yes, the new regulation, National Ambient Air Quality Standard is the policy change exploited for pollution awareness. However, before the NAAQS, PM2.5 monitoring was not mandatory.
  3. Results: we included the reference accordingly.
  4. Discussion and conclusion: we updated the discussion and conclusion session accordingly.
  5. Figures and tables:
    • We provided the direction, size and population of each city in the notes for Figure 1.
    • We provided the definition of urban and rural in the notes of Table 1.
    • We provided the definition of self-rated health, high vs low pollution in the notes of Table 3 and 4.

Reviewer 4 Report

Generally, there is a good discussion on limitations of the study and risks of sample bias etc.

I have one query which I couldn't find covered in the text or figures and tables:

I am assuming that the ground based observational readings are not be contemporaneous (taken at the same time) as the satellite derived data, and certainly not for PM2.5.  The limitation of this is that while the satellite data may give a good indication of total air borne particles, we cannot assume the composition at these time points matches the proportions of specific pollutants at ground level.  Is there any indication of the heterogeneity of the ground-based data?  Has the composition of PM2.5 particles changed over time and does it vary between urban and rural areas?

Page 3, line 90, states:

...the data allows us to partial out individual specific......charachteristics.

I am not sure whether this means 'parcel out' or whether it might be better to say 'separate out'.?

Author Response

Thank you very much for your comments. Our explanations and revisions are as follows:

  1. We agree with the concerns regarding satellite vs monitored PM2.5. The ground observation of PM2.5 for our study period is only partially available, as China mandated the observation and reporting of PM2.5 in stages from 2012 onwards. We acknowledge the limitation of not able to differentiate compositions of PM2.5. Following your suggestion, we point this out in our discussion of pollution data used. From our understanding, the change in composition over time is limited but it varies across urban and rural areas. This is discussed when introducing the pollution data in section 2.2.
  2. We have updated the language in describing individual fixed effects accordingly.